# Cardiac Cell and Animal Models for Duchenne Muscular Dystrophy in the Era of Gene Therapy and Precision Medicine

**DOI:** 10.3390/cells14171326

**Published:** 2025-08-27

**Authors:** Hidenori Moriyama, Toshifumi Yokota

**Affiliations:** 1Department of Medical Genetics, Faculty of Medicine and Dentistry, University of Alberta, Edmonton, AB T6G 2H7, Canada; hmoriyama@ualberta.ca; 2The Friends of Garrett Cumming Research & Muscular Dystrophy Canada HM Toupin Neurological Sciences Research, Edmonton, AB T6G 2H7, Canada

**Keywords:** Duchenne muscular dystrophy, dystrophin, gene therapy, precision medicine, cardiomyopathy, induced pluripotent stem cells, animal model

## Abstract

Duchenne muscular dystrophy (DMD) is a lethal inherited muscle disease caused by mutations in the *DMD* gene, and the development of gene therapies targeting *DMD* is rapidly progressing. Patient-derived induced pluripotent stem cells and animal models that mimic patient-specific mutations have significantly contributed to the advancement of precision medicine based on individual genetic profiles. Currently, no approved disease-specific therapy exists for DMD cardiomyopathy, which remains one of the leading causes of death in DMD patients. Therefore, the development of effective cardiac therapies represents a critical milestone in DMD research. In this review, we provide an overview of essential cellular and animal models used in DMD research, with a specific focus on the heart. We describe their key characteristics, advantages, and limitations. It is considered that a comprehensive and strategic integration of these models—based on a clear understanding of their respective strengths and weaknesses—will be important for advancing the development and clinical application of targeted therapies for DMD cardiomyopathy.

## 1. Introduction

Duchenne muscular dystrophy (DMD) is a fatal X-linked inherited muscular disorder caused by mutations in the *DMD* gene [1]. Dystrophin is a membrane-associated protein that stabilizes the sarcolemma by linking the actin cytoskeleton to the extracellular matrix [2,3]. The absence of dystrophin leads to progressive muscle degeneration and fibrosis. While progressive skeletal muscle weakness and loss of ambulation are hallmark features of DMD, most patients eventually die from respiratory failure or heart failure [4,5,6].

Recent advancements in therapeutic strategies targeting the genetic root cause of DMD have led to the emergence of gene therapy and precision medicine, which are now beginning to play central roles in clinical practice. Since 2016, four antisense oligonucleotides (ASOs) for exon-skipping have been approved by the FDA, and several others are currently under development [7]. Exon-skipping therapy is a promising approach that restores dystrophin protein production by modulating pre-mRNA splicing. By skipping specific exons, out-of-frame transcripts caused by deletion mutations can be converted into in-frame transcripts, resulting in the production of a truncated but partially functional dystrophin protein. This shifts the phenotype from severe DMD to the milder Becker muscular dystrophy (BMD) [1,8]. In 2023, micro-dystrophin replacement therapy also received its first FDA approval [9]. SRP-9001 (ELEVIDYS), developed by Sarepta Therapeutics, delivers micro-dystrophin cDNA under a muscle-specific promoter using an AAV vector, enabling functional protein expression in muscle tissue. Although pre-existing neutralizing antibodies to AAV can limit its use in some patients, micro-dystrophin therapy expands treatment options to a broader patient population. Additionally, various CRISPR-based gene editing strategies are rapidly progressing in preclinical and clinical studies [10].

While advancements in respiratory support and corticosteroid therapy have improved survival outcomes over time [6,11], many patients still develop dilated cardiomyopathy and die around their 30 s. This highlights the growing importance of managing and treating DMD cardiomyopathy. The clinical course of DMD cardiomyopathy is highly variable, ultimately progressing to congestive heart failure, arrhythmias, and sudden cardiac death [12,13]. Currently, there is no approved DMD-specific treatment for cardiomyopathy, although several new types of ASOs, such as peptide-conjugated ASOs, may provide potential benefits. Management relies primarily on standard heart failure medications such as β-blockers and Angiotensin-converting enzyme inhibitors/Angiotensin II receptor blockers.

Given this background, there is growing recognition of the need for therapeutic development specifically targeting DMD cardiomyopathy. Numerous cellular and animal models have been developed to study DMD, significantly contributing to therapeutic innovation. In this review, we focus on the cardiac aspects of DMD, providing an overview of the cellular and animal models used in DMD cardiac research. We describe their characteristics, advantages, and limitations, and introduce recent studies that have utilized these models to investigate gene therapy and precision medicine approaches. We also discuss how these models can be leveraged to advance the development and clinical translation of cardiac-targeted therapies in DMD.

## 2. Cell Models for DMD Research

### 2.1. Human Induced Pluripotent Stem Cell-Derived Cardiomyocytes (hiPSC-CMs)

Induced pluripotent stem cells (iPSCs) are powerful tools for disease modeling and drug discovery and have become a central cellular model in DMD research [14]. Human iPSCs (hiPSCs) share fundamental properties with embryonic stem cells, including the ability to differentiate into the trilaminar germ layers. With appropriate culture conditions and media, they can be directed to differentiate into various cell types [15,16]. As discussed in the following section, animal models of DMD have limitations in faithfully replicating the human disease phenotype, and therefore, access to patient-derived human cells represents a significant advantage. Indeed, cardiomyocytes derived from DMD patient iPSCs have greatly contributed to understanding the role of dystrophin in cardiomyocytes and to therapeutic screening [17]. These hiPSC-derived cardiomyocytes (hiPSC-CMs) exhibit a functional dystrophin–glycoprotein complex (DGC) localized to the membrane, show electrophysiological abnormalities and arrhythmogenicity, and reproduce phenotypes observed in patients with DMD cardiomyopathy [13,18,19]. Furthermore, comparative transcriptomic analyses of DMD hiPSC-CMs and left ventricular (LV) biopsy samples from patients have revealed shared molecular and biological pathway abnormalities [13].

For the development of gene therapy and precision medicine in DMD, it is essential to generate iPSC models that retain the genetic backgrounds of individual patients. These patient-specific hiPSC-CMs allow evaluation of the cardiac impact of specific *DMD* mutations and the therapeutic response following correction. For example, mutations that disrupt the actin-binding domain 1 (ABD-1), encoded by exons 2–8 of *DMD*, represent a hotspot accounting for approximately 7% of mutations in DMD patients [20]. Among the various mutations in this region, patients with deletions of exons 3–9 have been reported to present with mild or even asymptomatic phenotypes [21,22]. Kyrychenko and colleagues generated DMD hiPSC-CMs harboring mutations in the ABD-1 region and evaluated which gene-editing strategies would be most effective by assessing cardiomyocyte contractile function and calcium-handling properties. They demonstrated that CRISPR/Cas9-mediated deletion of exons 3–9 most effectively restored cardiomyocyte function, consistent with clinical observations [23].

In exon-skipping therapies, new strategies such as AI-based design tools have emerged to enhance efficiency and efficacy [24]. However, patient-derived iPSCs remain highly valuable. The reading-frame rule serves as the conceptual basis for exon skipping, where out-of-frame mutations typically lead to severe DMD, whereas in-frame mutations can produce truncated but partially functional dystrophin and result in the milder BMD phenotype [8,25]. Nonetheless, the genotype–phenotype relationship in DMD is more complex, and some in-frame deletions can still lead to severe symptoms [8,23,26,27]. Therefore, the functionality and stability of truncated dystrophin produced via exon skipping cannot be reliably predicted beforehand. Given the phenotypic variability in cardiac involvement, patient-derived hiPSC-CMs are indispensable tools for assessing the cardiac efficacy of such therapies.

There are known limitations of hiPSC-CMs. One major issue is their immature phenotype; compared to adult cardiomyocytes, they exhibit fetal-like characteristics in terms of structure, gene expression, conduction, ion channel profiles, and calcium handling kinetics [28]. In addition, DMD cardiomyopathy typically progresses over years to decades, but this long-term pathological progression is difficult to replicate in vitro. While these models do retain the patient’s genetic background, they cannot recapitulate systemic influences on the heart in a multisystem disease like DMD. The current approaches to generating patient-derived hiPSC models are also time-consuming and costly, which remains a major bottleneck in translating findings into patient care.

### 2.2. Advanced Cell Models for DMD—Three-Dimensional (3D) Cardiac Models

To overcome the structural and functional immaturity of hiPSC-CMs and to improve their utility for disease modeling, drug efficacy and toxicity testing, and mechanistic studies, numerous advanced strategies have been pursued [28]. Among them, the development of three-dimensional (3D) cardiac models using hiPSC-CMs has shown particular promise.

Organoids—3D cultures composed of multiple cell types—recapitulate key architectural and physiological features of native organs, including tissue-specific functions and cellular microenvironments [29]. These organoids possess self-organizing capabilities that allow for the reconstruction of tissue microarchitecture and matrix interactions. Cardiac organoids have been used to study the cellular components essential for heart development and to model disease states such as myocardial infarction and hypertrophic cardiomyopathy [30,31,32]. More recently, the generation of self-organizing chamber-like cardiac organoids, called “cardioids,” has been reported [33]. When cardiac organoids are generated from DMD patient-derived iPSCs, they allow the detection of transcriptional alterations and physiological or structural abnormalities in DMD cardiac tissue. Several of these features are not observable in traditional two-dimensional (2D) models [17,34,35,36]. A groundbreaking recent study reported the rapid generation of iPSCs from cryopreserved peripheral blood mononuclear cells and the use of organoid models to evaluate the effects of patient-specific ASOs in less than six weeks [37]. Given that the design and preclinical evaluation of ASOs are typically time- and cost-intensive (with preclinical studies in animal models often taking 6–8 years), this represents a significant advancement toward personalized ASO therapy. The study demonstrated restored dystrophin expression and normalization of rhythmic calcium fluctuations in the DMD cardiac organoids treated with ASOs. Additionally, even in organoids derived from patients with deep intronic variants, treatment with newly designed ASOs led to improvement in disease-associated phenotypes. These rapid and robust organoid-based platforms could significantly accelerate the realization of personalized ASO-based medicine.

Another promising approach combines hiPSC-CMs with bioengineering techniques [38]. In particular, 3D cell culture platforms known as engineered heart tissues (EHTs) provide anisotropic mechanical constraints on cardiomyocyte contraction, promoting the structural and functional maturation of hiPSC-CMs [39]. These more mature cardiac models are morphologically, functionally, and mechanically superior and are being used as platforms for detailed in vitro analyses and evaluation of gene therapies [38]. Notably, studies using 3D engineered heart muscle have demonstrated that CRISPR/Cas9-mediated correction of *DMD* mutations can restore dystrophin expression and improve cardiac contractility [40].

In summary, the use of patient-specific iPSCs and 3D cardiac models contributes to a deeper understanding of the fundamental mechanisms of DMD and facilitates the development of effective therapies. These models represent powerful tools with the potential to bridge the gap between basic research and clinical application in the era of next-generation genetic and personalized therapies. The challenge now lies in fully harnessing their potential to advance the development and implementation of genetic and precision therapies.

## 3. Animal Models for DMD Research

### 3.1. Non-Mammalian Models

The *DMD* gene is highly conserved across mammalian and non-mammalian species, making non-mammalian organisms useful for DMD research [41,42,43]. Commonly used non-mammalian models include the nematode *Caenorhabditis elegans*, the fruit fly *Drosophila melanogaster*, and the zebrafish *Danio rerio*. DMD models have been established in all three species and have served as valuable platforms for elucidating disease mechanisms and drug discovery. Compared to mammalian models, they are less subject to physiological complexity and individual variation, making them highly reproducible. These models also offer advantages such as ease of maintenance, short life cycles, large progeny numbers, and high amenability to genetic manipulation, which make them ideal for high-throughput drug screening [38,43]. In *Drosophila*, DMD models exhibit dilated cardiomyopathy-like phenotypes and reduced cardiac function [44]. Importantly, a promising drug delivery tool aimed at enhancing ASO delivery to the heart—a cell-penetrating peptide named DG9—was identified through screening studies using zebrafish models [45,46].

### 3.2. Mouse Models

#### 3.2.1. *mdx* Mice

The *mdx* mouse is the most widely studied and historically established DMD model, carrying a nonsense mutation in exon 23 of the *DMD* gene that leads to dystrophin deficiency [47,48,49]. These mice begin to show skeletal muscle damage around three weeks of age and develop progressive fibrosis, fatty infiltration, and muscle atrophy in most skeletal muscles. Because *mdx* mice have a normal to slightly shortened lifespan (~20% reduction), remain fertile, and are relatively easy to maintain, they offer several experimental advantages including extensive natural history data and established protocols [43,50,51,52]. A majority of preclinical studies for DMD therapies—including those that have advanced to clinical trials—have used *mdx* mice [53].

One major limitation of the *mdx* mouse model is its relatively mild phenotype. In particular, the cardiac phenotype is subtle, which can be a constraint when studying DMD cardiomyopathy. Indicators of cardiac dysfunction such as reduced stroke volume and cardiac output become prominent only after one year of age, and these changes are generally more pronounced in female mice [54,55,56]. Cardiac fibrosis is typically observed later in life. Studies have shown that as little as 4–15% dystrophin expression in cardiomyocytes can delay or ameliorate cardiomyopathy, while expression in ~50% of cardiomyocytes can prevent it [55,57].

Interestingly, metabolic abnormalities and changes in signaling pathways have been detected in *mdx* hearts as early as 3 months of age—before structural abnormalities are detectable by histology or echocardiography [58]. To unmask the underlying cardiac vulnerability in *mdx* mice, researchers have employed stress-inducing protocols such as dobutamine or isoproterenol administration and aortic constriction [59,60,61]. These studies revealed that dystrophin-deficient hearts are abnormally sensitive to mechanical stress and load-induced injury [61], highlighting the essential role of dystrophin in protecting cardiomyocytes against such mechanical insults [62].

The *mdx* mouse model has also been widely used to evaluate cardiac outcomes in gene therapy studies. Micro- or mini-dystrophin therapy uses AAV vectors to deliver shortened versions of the *DMD* gene to skeletal and cardiac muscle. Several studies have demonstrated that administration or cardiac-specific expression of micro-dystrophin restores DGC localization and sarcolemmal stability, attenuates age-dependent cardiac fibrosis, and prevents functional decline [63,64,65]. A single administration of micro-dystrophin results in sustained expression in cardiomyocytes and improves hemodynamic parameters [66]. While the benefits are limited in late-stage dilated cardiomyopathy, improvements in stress tolerance (e.g., under dobutamine challenge) have been reported [67]. Using *mdx* mice to evaluate different treatment timings and follow-up durations provides critical insights for patient selection and clinical trial design.

Because cardiac phenotypes emerge later in *mdx* mice, they are often used for long-term evaluation of genome editing therapies. For example, Hakim et al. conducted a long-term study using AAV-mediated CRISPR/Cas9 genome editing in *mdx* mice [68]. Intravenous delivery of AAV vectors encoding Cas9 and dual gRNAs targeting introns 22 and 23 resulted in dystrophin restoration in the heart 12 months post-injection, along with reduced cardiac fibrosis and improved ejection fraction. Similarly, another study showed improved stroke volume and cardiac output at 19 months following neonatal genome editing using AAV-CRISPR/Cas9 vectors [69].

Additionally, rAAVrh74.MCK.GALGT2, which underwent Phase I/IIa clinical trials [70], was preclinically evaluated in *mdx* mice. *GALGT2* encodes a glycosyltransferase that enhances the expression of several proteins with therapeutic potential, including utrophin, plectin1, agrin, laminin α4/α5, and integrins α7/β1 [71,72,73]. GALGT2 expression is normally low in the heart, but *GALGT2*-deficient *mdx* mice display worsened cardiac pathology and reduced function compared to *mdx* controls [74,75]. Xu et al. showed that neonatal delivery of rAAVrh74.MCK.GALGT2 suppressed early LV remodeling and fibrosis-related gene expression without negatively impacting hemodynamics at 3 months of age [73]. Moreover, systemic administration at 2 months of age, followed by evaluation at 17 months of age, revealed significantly improved stroke volume and cardiac output both at rest and under dobutamine challenge.

#### 3.2.2. Modified *mdx* Mouse Models

To overcome the relatively mild phenotype of *mdx* mice, various modified models have been developed to more closely replicate the dystrophic phenotype seen in human DMD, particularly with more pronounced cardiac involvement [50,76]. One strategy involves backcrossing *mdx* mice (originally on the C57BL/10 background) with other mouse strains to induce phenotypic variation [50]. For example, when *mdx* mice are backcrossed onto a DBA/2J background, the resulting mice exhibit exacerbated dystrophic features, including increased fibrosis and reduced muscle regeneration [77,78]. In this strain, cardiac fibrosis and calcification appear as early as 10 weeks of age, accompanied by elevated expression of genes related to fibrosis and cardiac function such as *Col1a1* and *Nppa* [78]. However, other study reports that even non-DMD control DBA/2J mice show cardiac abnormalities and hemodynamic changes, casting doubt on the suitability of this background for modeling cardiomyopathy [79].

One reason for the mild phenotype of *mdx* mice is believed to be compensatory upregulation of utrophin, a structural protein homologous to dystrophin [80]. Utrophin shares a high amino acid sequence similarity with dystrophin and plays a similar role in linking the sarcolemma to the cytoskeleton. In utrophin-deficient *mdx* mice (*mdx/utrn* double knockout mice), a much more severe phenotype is observed, including growth retardation, weight loss, spinal curvature, and premature death due to respiratory failure at around 20 weeks [81]. Histological analysis of hearts from 8–11 week-old double knockout mice revealed inflammatory cell infiltration and cardiomyocyte necrosis, although ventricular hypertrophy or dilation was not initially observed [82]. Later studies reported progressive deterioration of cardiac function, including reduced fractional shortening and ejection fraction, increased end-diastolic volume, ventricular dilation, and wall thinning by 15 weeks of age [83]. A related model known as Fiona/dko mice—dystrophin/utrophin double knockout mice expressing a human utrophin transgene specifically in skeletal muscle—was created to isolate and study cardiac progression independently. Using this model, micro-dystrophin therapy was shown to delay the onset of heart failure and suppress cardiac functional and pathological deterioration, including strain imaging markers [84,85]. Other modified *mdx* mouse models include those lacking proteins related to the sarcolemma or cytoskeleton, such as α7-integrin [86] and α-dystrobrevin [87], which also help elucidate individual molecular pathways in DMD.

Additional models have been generated by impairing muscle regeneration or mimicking human genetic features. For example, *mdx* mice lacking MyoD, a master regulator of muscle differentiation, or telomerase RNA (mTR) to inhibit telomere maintenance, exhibit more severe dystrophic phenotypes, including worsened cardiac fibrosis and reduced cardiac function [88,89]. The *mdx* mice with targeted deletion of *Cmah*, a gene inactivated in humans, also display a more severe dystrophic phenotype with relevance to human pathology [90].

These double knockout models allow for faster disease progression, making it possible to detect treatment effects—including improvement in cardiomyopathy or survival benefits—within a shorter timeframe. However, they also present challenges. The deletion of additional genes beyond dystrophin complicates data interpretation, the models are often difficult to breed and maintain, and comprehensive natural history data are still not enough. As a result, their use in preclinical research remains limited [43,50].

#### 3.2.3. DMD Mouse Models for Personalized Medicine

In the development of exon-skipping therapies and genome editing, it is essential to test therapeutic strategies in animal models that reflect patient-specific genetic mutations. Over 7000 distinct mutations in the *DMD* gene have been identified in patients [8,91], making it impractical to generate mouse models for all the mutations. Therefore, models have primarily focused on common mutation hotspots, such as the exon 44–55 and exon 2–20 regions, enabling validation of sequence-specific, personalized therapies applicable to broader patient populations.

Atmanli et al. used a DMD mouse model lacking exon 44 (ΔEx × 44) to investigate the cardiac effects of CRISPR/Cas9-mediated dystrophin restoration targeting exon 45 [92]. After intraperitoneal injection of AAV vectors carrying Cas9 and gRNAs on postnatal day 4, the authors evaluated cardiac tissues at 18–22 months of age and found histological and transcriptomic improvements associated with dystrophin restoration.

A *Dmd* Δ52–54 mouse model, in which exons 52 to 54 are deleted, has also been reported to display cardiac phenotypes more closely resembling DMD cardiomyopathy. From 12 weeks of age, these mice exhibit increased cell surface area of cardiomyocytes, concentric LV hypertrophy (characterized by increased wall thickness and reduced end-systolic diameter), elevated ejection fraction, and tachycardia—all of which persist through at least 52 weeks of age [93]. Using this model, CRISPR/Cas9-mediated exon 55 skipping via a single-cut strategy during the neonatal period was shown to prevent the cardiomyopathic phenotype. By 12 weeks of age, treated mice demonstrated improved heart rate, reduced ventricular wall thickness, and decreased fractional shortening [94]. Because this model presents early-onset cardiac features, it serves as a valuable platform for developing therapies specifically targeting cardiac involvement in DMD [95].

To overcome the species-specific differences between mouse and human genomic sequences, humanized DMD mouse models have also been developed. These transgenic mice carry the entire human *DMD* gene (h*DMD*) or lack murine dystrophin expression while harboring the human *DMD* gene (h*DMD*/*mdx*). Using these models, targeted mutations have been introduced into human sequence hotspot regions to mimic patient-specific genotypes [96,97]. These humanized models are highly suited for in vivo validation of sequence-specific ASOs and CRISPR-based therapies designed for human patients and have been used to assess therapeutic efficacy in a context that more closely replicates genetic conditions [97,98,99].

### 3.3. Rat Models

Compared to mice, rats are approximately ten times larger in body size, yet they are easier to handle and less expensive to maintain than large animal models. Using TALEN-mediated genome editing, researchers generated a *Dmd^mdx^* rat model by targeting exon 23 of the *DMD* gene [100]. These rats display muscle fatigue and elevated serum creatine kinase (CK) levels. Histologically, severe muscle fiber necrosis and regeneration are observed in limb and diaphragm muscles as early as 3 months of age, with marked fibrosis and fatty infiltration evident by 7 months. At 3 months, echocardiography reveals concentric LV hypertrophy and diastolic dysfunction, although systolic function remains preserved. In more recent studies using this model, significant reductions in LV ejection fraction and ventricular dilation were observed from 3 months of age onward. By 7–9 months, signs of diastolic dysfunction, increased LV filling pressures, and pulmonary hypertension became apparent [101]. Invasive hemodynamic assessments confirmed decreased LV contractility and elevated LV filling pressures. Histological analysis at 9 months showed prominent cardiac fibrosis accompanied by increased inflammation and oxidative stress. A separate rat DMD model was generated using CRISPR/Cas9 technology targeting exons 3 and 16 of the *DMD* gene [102]. While no significant cardiac changes were observed at 6 months of age, reduced LV systolic function and prominent fibrosis were evident by 10 months of age [103].

The *Dmd^mdx^* rat model was also employed in the preclinical evaluation of fordadistrogene movaparvovec, an rAAV9-based mini-dystrophin therapy (now discontinued), to determine pharmacologically effective dosing [104]. Treatment restored dystrophin expression in both skeletal and cardiac muscle and improved cardiac remodeling as assessed by echocardiography. Histological analyses confirmed attenuation of cardiac fibrosis. Notably, therapeutic benefits were also observed when the treatment was administered to older rats (4–6 months), suggesting potential utility even in the progressive stages of DMD.

Compared to *mdx* mice and canine models, rats exhibit an earlier onset of cardiac pathology, which makes them particularly advantageous for preclinical testing of cardiac therapies. Although the diversity of available mutations in rat models is currently limited—posing challenges for precision medicine applications—the increasing accessibility of genome editing tools like CRISPR offers opportunities for the development of customized rat models. In the near future, humanized DMD rat models may play a pivotal role in translational research and drug development [105].

### 3.4. Rabbit Models

Rabbits share closer physiological, anatomical, and genetic characteristics with humans than mice, while still offering advantages over large animals, such as lower maintenance costs and shorter gestation periods [106]. Using CRISPR/Cas9-based genome editing with a guide RNA targeting exon 51 of the *DMD* gene, Sui et al. successfully generated a DMD rabbit model [107]. This DMD rabbit exhibits hallmark features of the disease, including severe motor dysfunction, elevated serum CK levels, and progressive skeletal muscle necrosis and fibrosis. Notably, by 5 months of age, clear pathological changes were observed not only in the diaphragm but also in the heart, demonstrating similarities to human DMD pathology. Echocardiographic assessments revealed ventricular dilation and reduced ejection fraction and fractional shortening. Although current findings remain limited, the DMD rabbit model shows great promise as a platform for studying DMD-associated cardiomyopathy.

### 3.5. Canine Models

Canine DMD models are widely accepted as clinically relevant large animal models, closely recapitulating the human disease phenotype. While cases of dystrophinopathy have been reported in various dog breeds [50], two models—Golden Retriever Muscular Dystrophy (GRMD) and Canine X-linked Muscular Dystrophy (CXMD)—are the most extensively studied worldwide.

GRMD dogs, first identified in the 1980s, carry a spontaneous mutation in the *DMD* gene and represent the first established experimental colony of DMD dogs. These animals exhibit elevated serum CK, muscle pathology, and clinical symptoms closely resembling those of human DMD patients [50,108,109]. The causative mutation is a nonsense variant near the end of intron 6, in which an A-to-G substitution disrupts the conserved splice acceptor site, leading to aberrant splicing from exon 6 to exon 8 and subsequent frameshift. Due to this mutational feature, restoration of an in-frame transcript requires multi-exon skipping, making GRMD a valuable model for studying such therapeutic strategies [110,111]. A longitudinal study of cardiac function in GRMD demonstrated a significant reduction in LV ejection fraction beginning at 9 months of age, with values falling below 50% by 24 months [112]. Speckle-tracking echocardiography revealed abnormalities in radial systolic myocardial velocity gradient, LV twist, and longitudinal strain as early as 2 months of age, suggesting these indices may serve as early biomarkers of myocardial involvement in DMD. The utility of speckle-tracking echocardiography for drug efficacy evaluation has also been reported [113], and its standardization in preclinical research is expected to progress in parallel with clinical use. Conversely, a separate study using echocardiography and cardiac MRI to evaluate adult GRMD dogs reported the onset of LV systolic dysfunction between 30 and 45 months. Nevertheless, even in that study, speckle-tracking echocardiography-derived circumferential strain proved useful for early detection of cardiac dysfunction [114]. Late gadolinium enhancement on cardiac MRI in GRMD revealed lesions in the lateral wall of the left ventricle, similar to those observed in DMD patients, as well as earlier involvement of the anterior septum.

The CXMD model was established by artificial insemination of beagles using sperm from GRMD dogs. These dogs harbor the same *DMD* gene mutation as GRMD [115]. While CXMD dogs exhibit similar clinical and pathological findings [116], they are smaller and present with a milder phenotype, making colony maintenance and handling more feasible. Cardiac pathology also progresses more slowly and mildly compared to GRMD. Electrocardiogram abnormalities, particularly the emergence of Q waves, become evident around 6–7 months of age; however, concomitant reductions in LV function on 2D echocardiography are not clearly observed. Histologically, among eight dogs evaluated at 21 months of age, three exhibited definitive LV fibrosis [117]. At a mean age of 24 months, no marked ventricular dilation or systolic dysfunction was detected on echocardiography. Only the peak radial strain rate during early diastole in the posterior segment, assessed by 2D speckle-tracking echocardiography, suggested early myocardial impairment [118]. Interestingly, Purkinje fibers show vacuolar degeneration as early as 4 months of age—even in the absence of overt ventricular pathology—raising the possibility that such changes contribute to the characteristic Q waves and life-threatening arrhythmias seen in DMD [119]. Echigoya et al. demonstrated that an ASO cocktail therapy successfully restored dystrophin expression in Purkinje fibers, reduced vacuolar degeneration, and alleviated conduction defects [120]. These findings suggest that CXMD may be particularly suitable for evaluating cardiac conduction abnormalities and arrhythmias, rather than systolic function alone.

The ΔE50-MD canine model carries a missense mutation at the 5′ splice site of intron 50 in the *DMD* gene, resulting in out-of-frame skipping of exon 50 [121,122]. This mutation was first identified as a naturally occurring variant in the Cavalier King Charles Spaniel and has since been maintained on a beagle background. Due to the nature of this mutation, the model is suitable for the development of gene therapies targeting hotspot mutations in the *DMD* gene. Using this model, Amoasii et al. tested a single-cut genome editing strategy employing AAV9-CRISPR/Cas9 delivery [123]. Six weeks after systemic administration, dystrophin expression in the heart reached up to 92% of normal levels. However, this study did not report the impact of the therapy on cardiac function or histopathology.

Because of their large body size relative to mice, dogs are attractive models for scaling up gene and cell therapies, and they can provide valuable translational insights [43]. However, as with other large animals, colony maintenance is extremely costly and logistically demanding, restricting research to a limited number of institutions. Moreover, considerable inter-individual variability in phenotype poses challenges for statistical analysis [124,125,126]. Cardiac dysfunction and structural abnormalities also emerge relatively late in the disease course, necessitating long-term maintenance. Thus, the use of early-detection imaging modalities such as speckle-tracking echocardiography is critical to capturing cardiac phenotypes relevant to gene therapy development.

### 3.6. Pig Models

Pigs are considered a promising large animal model for DMD due to their genetic, physiological, and anatomical similarities to humans—particularly with regard to cardiac structure and function. A DMD pig model (*DMD*Δexon52) was generated by targeted deletion of exon 52 in the *DMD* gene in male porcine cells, followed by somatic cell nuclear transfer to produce offspring [127]. Although potential epigenetic alterations associated with somatic cell nuclear transfer should be taken into consideration, these pigs exhibited complete loss of dystrophin in skeletal muscle, elevated serum CK levels, progressive muscular dystrophy, impaired motor function and muscle strength, and ultimately died from respiratory failure within a maximum lifespan of approximately three months. Comprehensive transcriptomic analyses of cardiac tissue have not yet been reported, and future studies are awaited. Moretti et al. used the *DMD*Δexon52 pig model to evaluate CRISPR/Cas9-based genome editing for reading frame restoration [128]. They systemically delivered AAV9 vectors carrying two gRNAs targeting intronic regions flanking exon 51 and assessed the animals 3–4 months later. Restoration of dystrophin expression was confirmed in both skeletal and cardiac muscle. In addition, they performed detailed cardiac evaluations, including LV angiography via catheterization and electrophysiological mapping. Although AAV treatment did not improve LV systolic or diastolic function, high-dose therapy reduced the extent of low-voltage areas, and autopsy findings showed attenuation of fibrosis progression. Autopsy and video analysis further suggested that the primary cause of death in *DMD*Δexon52 pigs was not heart failure but rather fatal arrhythmias. Meanwhile, Stirm et al. recently generated a model of ubiquitous correction of DMD by systemic deletion of exon 51 in *DMD*Δexon52 pig model and analyzed its phenotype [129]. The resulting *DMD*Δ51–52 model did not show the characteristic dystrophic alterations observed in *DMD*Δexon52 pigs, and exhibited normalization of LVEF and the myocardial proteome profile. These findings support the validity of the exon-skipping strategy and highlight the utility of this model as a mild BMD pig model.

This pig model enables highly detailed assessments of cardiac function, hemodynamics, and electrophysiology under conditions that closely mimic clinical settings. However, early mortality remains a major limitation, hindering long-term evaluation of disease progression and therapeutic interventions. Additional constraints include the long gestation period of pigs (~114 days), the need for large housing facilities and high maintenance costs, and strict regulations regarding the use of genetically modified pigs in research [10].

To overcome high neonatal mortality, a DMD model using the smallest breed of pig, the microminipig, has also been developed. Using CRISPR/Cas9 technology, a model carrying a targeted mutation in exon 23 of the *DMD* gene was generated [130]. These pigs exhibited clear clinical phenotypes, including motor abnormalities, generalized skeletal muscle weakness and atrophy, and markedly elevated serum CK levels. The maximum lifespan was reported as 29.9 months. Although only a small number of animals have been analyzed, echocardiography showed a reduction in ejection fraction from as early as six months of age, with progressive decline over time. Histological analysis at 12 months revealed significant cardiac fibrosis. This model holds promise as a valuable tool for studying DMD-associated cardiomyopathy in a clinically relevant context.

### 3.7. Monkey Models

Non-human primates are evolutionarily the closest species to humans and share high degrees of similarity in genetics, fine motor function, muscle structure and physiology, as well as immune and metabolic systems [131]. Chen et al. generated chimeric rhesus monkeys by targeting exon 5 of the *DMD* gene using CRISPR/Cas9 gene editing [132]. Building upon this chimera, the same group subsequently established a DMD monkey model with complete dystrophin deficiency. These animals exhibited progressive muscle degeneration and motor dysfunction that closely mirrored the human disease phenotype [131]. Single-cell RNA sequencing of skeletal muscle from the DMD monkeys revealed a dramatic increase in immune cell infiltration, alterations in fibro-adipogenic progenitor populations, and impaired differentiation of muscle stem cells. At 1 year of age, echocardiographic analysis did not identify significant abnormalities in cardiac structure or function. This is consistent with the clinical course of DMD in humans, where cardiac manifestations often appear at later stages. However, how the DMD monkey model can be best leveraged for cardiac therapeutic development remains an open question.

In summary, a wide range of animal models applicable to DMD research has been developed, contributing significantly to the understanding of disease mechanisms and the advancement of novel therapeutic approaches. In vivo studies targeting DMD cardiomyopathy are also progressing; however, the severity and onset of cardiomyopathy vary across models, each possessing distinct advantages and limitations (Table 1). Detailed knowledge of the cardiac phenotypes in these models remains incomplete. Continued accumulation of such data is expected to support the development of higher-quality research for effective cardiac therapies in DMD.

## 4. Conclusions and Future Perspectives

In recent years, the development of gene therapies for DMD has advanced rapidly, ushering in an era of precision medicine that accounts for each patient’s unique genetic background. To date, clinical trials for *DMD* gene therapies have primarily focused on evaluating dystrophin restoration and improvements in skeletal muscle symptoms. In contrast, endpoints related to cardiac involvement have largely been absent. Common inclusion/exclusion criteria regarding cardiac function have relied on echocardiographic measurements, typically using a cutoff value for LV ejection fraction of 40% or 50% at screening, with a requirement that participants be in a stable clinical state [133,134,135,136,137]. Moreover, due to concerns over arrhythmia risk, many trials have excluded individuals with QTc intervals exceeding 450 ms. The long-term cardiac outcomes of gene therapy remain largely unknown, and careful monitoring is essential to determine how these therapies will ultimately impact the heart.

Advances in technologies such as iPSCs and CRISPR-based genome editing have facilitated the development of diverse cardiac in vitro and in vivo models, greatly contributing to the research and development of gene and precision therapies. hiPSC-CMs from DMD patients have demonstrated the ability to replicate key molecular and functional features of human cardiac tissue. Their 3D culture systems further enhance physiological relevance by mimicking native microenvironments. In vivo, the creation of mouse models that carry patient-specific *DMD* mutations—including humanized models—has enabled precise therapeutic validation tailored to individual genotypes. In addition, insights are steadily accumulating from studies evaluating cardiac phenotypes and therapeutic responses in large animal DMD models.

Despite these advances, no single model fully recapitulates the complexity of DMD cardiomyopathy. Even models that share the same genetic mutation as patients may not faithfully reproduce the disease’s clinical course, particularly its heterogeneous cardiac progression. Therefore, a nuanced understanding of each model’s strengths and limitations is essential. Researchers must match model selection to study objectives, acknowledge inherent limitations, and integrate findings across multiple systems to build a comprehensive view. Furthermore, leveraging emerging technologies for advanced analysis will be crucial. In the era of precision medicine, the establishment of high-quality databases that integrate genetic profiles with detailed cardiac clinical data will also be indispensable. Ultimately, by effectively utilizing available cell and animal models, the development and clinical testing of therapies specifically targeting DMD cardiomyopathy are expected to accelerate, with the hope that new treatments will reach patients in the near future (Figure 1).

## Figures and Tables

**Figure 1 cells-14-01326-f001:**
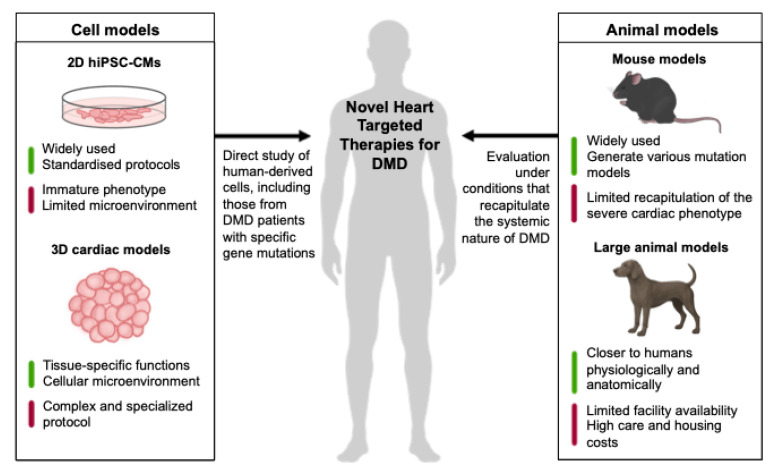
Cell and animal cardiac models for developing novel heart-targeted therapies for DMD.

**Table 1 cells-14-01326-t001:** Comparison of representative animal models for DMD cardiomyopathy for mechanistic studies and therapeutic development.

Species	Model	Mutation	Cardio-Myopathy	Advantages	Limitations
Mouse	*mdx*	Exon 23	Mild	Widely used, Extensive knowledge	Late onset cardiac phenotype (>10–12 months)
Mouse	DBA/2J *mdx*	Exon 23	Present	More severe than *mdx* mice	Cardiac calcification (rare in patients), Possible cardiac abnormality in non-*mdx* mice
Mouse	*mdx/utrn* KO	*DMD + utrophin*	Severe	Marked phenotype, Shorter lifespan	Utrophin deficiency
Mouse	*DMD*Δ52–54	Exon 52–54	Present	Mimics human hotspot mutation	Limited availability
Rat	*DMD^mdx^* rat	Exon 23	Present	Intermediate size, Early cardiac involvement (3 months)	Limited model diversity and insufficient data
Rabbit	*DMD* KO rabbit	Exon 51	Present	Intermediate size	Limited model diversity and insufficient data
Canine	GRMD	Intron 6	Present	Closer to human patients than small animal models	High maintenance and housing costs
Canine	CXMD	Intron 6	Mild	Small and easy to handle	Late onset of systolic dysfunction
Pig	*DMD*Δ52	Exon 52	Present	Heart structure and function similar to humans	High neonatal mortality and early death
Monkey	DMD monkey	Exon 5	Unknown	Non-human primate	No evidence of cardiomyopathy

KO, knockout; GRMD, Golden Retriever Muscular Dystrophy; CXMD; Canine X-linked Muscular Dystrophy.

## Data Availability

No new data were created or analyzed in this study. Data sharing is not applicable to this article.

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
