# Peer review of "Cardiac Cell and Animal Models for Duchenne Muscular Dystrophy in the Era of Gene Therapy and Precision Medicine"

_cells, 2025, doi:10.3390/cells14171326_

Round 1
Reviewer 1 Report
Comments and Suggestions for Authors
In this manuscript, Moriyama and Yokota provide an overview of essential cellular and animal models used in duchenne muscular dystrophy(DMD) research. They systemically describe the key characteristics, advantages, and limitations of different models, which may significantly improve the understanding of DMD and the development of novel therapeutic strategies. However, there are a few problems needed to be addressed in the revision. I will list them as follows:
- In general, the studies cited for each model are classical and significant. However, some new studies (PMID: 37428903, et al) have not been mentioned in current manuscript. Authors may need to make a balance in the citation of classical studies and novel studies.
- The Table used for representing animal models for DMD cardiomyopathy is well generated and easy to understand. The only improvement needed is to add one column descripting whether those models are utilized for therapeutics or only for mechanistic studies. In addition, it will be helpful to add relative citation number for each model in this table.
Author Response
Reviewer 1
In this manuscript, Moriyama and Yokota provide an overview of essential cellular and animal models used in duchenne muscular dystrophy(DMD) research. They systemically describe the key characteristics, advantages, and limitations of different models, which may significantly improve the understanding of DMD and the development of novel therapeutic strategies. However, there are a few problems needed to be addressed in the revision. I will list them as follows:
We thank the reviewer for the constructive comments. In the revised manuscript, we have modified the manuscript that addressed the concerns of the reviewer.
- In general, the studies cited for each model are classical and significant. However, some new studies (PMID: 37428903, et al) have not been mentioned in current manuscript. Authors may need to make a balance in the citation of classical studies and novel studies.
Thank you for your valuable suggestion and for providing the important previous study. To provide readers with more up-to-date insights, we have added the following content.
Meanwhile, Stirm et al. recently generated a model of ubiquitous correction of DMD by systemic deletion of exon 51 in DMDΔexon52 pig model and analyzed its phenotype [129]. The resulting DMDΔ51-52 model did not show the characteristic dystrophic alterations observed in DMDΔexon52 pigs, and exhibited normalization of LVEF and the myocardial proteome profile. These findings support the validity of the exon-skipping strategy and highlight the utility of this model as a mild BMD pig model.
- The Table used for representing animal models for DMD cardiomyopathy is well generated and easy to understand. The only improvement needed is to add one column descripting whether those models are utilized for therapeutics or only for mechanistic studies. In addition, it will be helpful to add relative citation number for each model in this table.
We believe that all of these models are useful for both mechanistic studies and therapeutic development, serving as a foundation for DMD research. Therefore, we have revised the table title to “Comparison of representative animal models for DMD cardiomyopathy for mechanistic studies and therapeutic development.” Also, relative citation numbers are certainly a useful indicator; however, as the purpose of this table is to summarize key features, we chose not to include them. We nevertheless appreciate this valuable suggestion.
Reviewer 2 Report
Comments and Suggestions for Authors
In the manuscript titled "Cardiac Cell and Animal Models for Duchenne Muscular Dystrophy in the Era of Gene Therapy and Precision Medicine" by Moriyama H. and Yokota T., the authors review models for DMD with a focus on cardiac function. Overall, the review is well-written, clear, concise, and logically organized. The discussion begins with an overview of 2D and 3D cellular models, followed by various animal models.
I do not have any major issues; however, I have a few minor issues:
1. The authors state, "Currently, no disease-specific therapy is available for DMD cardiomyopathy, which remains one of the leading causes of death in DMD patients." Considering recent advances—such as gene therapy using SRP-9001 and antisense oligonucleotides (ASO)—doesn't evidence suggest that these approaches improve cardiomyopathy symptoms? As noted in the article by Zaidman et al. (Ann Neurol, 2023), SRP-9001 has been shown to deliver the therapeutic gene into cardiomyocytes. It would be worthwhile to discuss whether gene therapy (e.g., SRP-9001) or pathogenetic therapy (e.g., ASO) contributes to reductions in cardiomyopathy severity.
2. Additionally, illustration appears as a simple blue rectangle, likely due to its format or resolution.
With these revisions, I believe the manuscript can be suitable for acceptance and publication in the Cells journal.
Author Response
Reviewer 2
In the manuscript titled "Cardiac Cell and Animal Models for Duchenne Muscular Dystrophy in the Era of Gene Therapy and Precision Medicine" by Moriyama H. and Yokota T., the authors review models for DMD with a focus on cardiac function. Overall, the review is well-written, clear, concise, and logically organized. The discussion begins with an overview of 2D and 3D cellular models, followed by various animal models.
I do not have any major issues; however, I have a few minor issues:
We thank the reviewer for the constructive comments. In the revised manuscript, we have modified the manuscript that should satisfactorily address the concerns of the reviewers.
- The authors state, "Currently, no disease-specific therapy is available for DMD cardiomyopathy, which remains one of the leading causes of death in DMD patients." Considering recent advances—such as gene therapy using SRP-9001 and antisense oligonucleotides (ASO)—doesn't evidence suggest that these approaches improve cardiomyopathy symptoms? As noted in the article by Zaidman et al. (Ann Neurol, 2023), SRP-9001 has been shown to deliver the therapeutic gene into cardiomyocytes. It would be worthwhile to discuss whether gene therapy (e.g., SRP-9001) or pathogenetic therapy (e.g., ASO) contributes to reductions in cardiomyopathy severity.
We thank the reviewer for this valuable comment. We agree that SRP-9001 (microdystrophin therapy) and ASOs, particularly new types such as peptide-conjugated ASOs, can exert effects on the myocardium and potentially improve cardiomyopathy. To provide a more accurate statement, we have revised the text as follows.
Abstract: Currently, no approved disease-specific therapy exists for DMD cardiomyopathy, which remains one of the leading causes of death in DMD patients.
Introduction: Currently, there is no approved DMD-specific treatment for cardiomyopathy, although several new types of ASOs, such as peptide-conjugated ASOs, might provide potential benefits. Management relies primarily on standard heart failure medications such as β-blockers and Angiotensin-converting enzyme inhibitors/Angiotensin II receptor blockers.
- Additionally, illustration appears as a simple blue rectangle, likely due to its format or resolution.
We apologize for the inappropriate image. It has been replaced with the correct one.
Reviewer 3 Report
Comments and Suggestions for Authors
Thanks to the authors for this interesting review. It is well organized and written. The mastership of the authors on both subjects Genomics and cardiac phenotype evaluation in such a wide spread of species is very valuable
Therefore I have only few critics. In order in apparition in the text :
Minor - Lines 90-95 merit to be clarified
Minor - Lines 96-99 there is redundancy in the sentences that must be corrected
Major - Line 421. The problem of interindividual variability in phenotypes is only discussed through reference (124). This is very damageable, WGS and transcriptomics for the study of this interindividual variability is of major importance for the study of alternative pathways and emergencies of new hypothesis. Even in cells culture and animal models
Major – Line 431. Somatic cell nuclear transfer to produce offsprings leads also to epigenetic alterations. A comparative careful parallel analysis of the controls must be done. Please specify, if this has been done and sharpens the limitation of the model.
In conclusion, congratulations, your didactic review will be an help for the scientific community (especially the youngests) involved in this great therapeutic adventure.
Author Response
Reviewer 3
Thanks to the authors for this interesting review. It is well organized and written. The mastership of the authors on both subjects Genomics and cardiac phenotype evaluation in such a wide spread of species is very valuable
Therefore I have only few critics. In order in apparition in the text :
We thank the reviewer for the constructive comments. In the revised manuscript, we have modified the manuscript that should satisfactorily address the concerns of the reviewers.
Minor - Lines 90-95 merit to be clarified
We thank the reviewer for this valuable comment. We have revised the text as below to make it easier for readers to understand.
For example, mutations that disrupt the actin-binding domain 1 (ABD-1), encoded by exons 2–8 of DMD, represent a hotspot accounting for approximately 7% of mutations in DMD patients [20]. Among the various mutations in this region, patients with deletions of exons 3–9 have been reported to present with mild or even asymptomatic phenotypes [21, 22]. Kyrychenko and colleagues generated DMD hiPSC-CMs harboring mutations in the ABD-1 region and evaluated which gene-editing strategies would be most effective by assessing cardiomyocyte contractile function and calcium-handling properties. They demonstrated that CRISPR/Cas9-mediated deletion of exons 3–9 most effectively restored cardiomyocyte function, consistent with clinical observations [23].
Minor - Lines 96-99 there is redundancy in the sentences that must be corrected
Thank you for your valuable suggestion. We have revised the sentences to correct them.
Major - Line 421. The problem of interindividual variability in phenotypes is only discussed through reference (124). This is very damageable, WGS and transcriptomics for the study of this interindividual variability is of major importance for the study of alternative pathways and emergencies of new hypothesis. Even in cells culture and animal models
Thank you for your valuable suggestion. We have added the relevant references.
Major – Line 431. Somatic cell nuclear transfer to produce offsprings leads also to epigenetic alterations. A comparative careful parallel analysis of the controls must be done. Please specify, if this has been done and sharpens the limitation of the model.
Thank you for your valuable suggestion. In this study, a pig model was generated using somatic cell nuclear transfer, and genome-wide transcriptome analyses were performed on biceps femoris muscle samples from 2-day-old and 3-month-old DMD pigs, as well as age-matched wild-type pigs. Transcriptome changes in 3-month-old DMD pigs closely matched the gene expression profiles observed in human DMD, reflecting processes such as muscle degeneration, regeneration, inflammation, fibrosis, and reduced metabolic activity. In contrast, the transcriptome profile of 2-day-old DMD pigs resembled changes induced by acute exercise-induced muscle injury. Comprehensive analyses of cardiac tissue have not yet been conducted. Based on these findings, we have revised the main text as follows.
Although potential epigenetic alterations associated with somatic cell nuclear transfer should be taken into consideration, these pigs exhibited complete loss of dystrophin in skeletal muscle, elevated serum CK levels, progressive muscular dystrophy, impaired motor function and muscle strength, and ultimately died from respiratory failure within a maximum lifespan of approximately three months. Comprehensive transcriptomic analyses of cardiac tissue have not yet been reported, and future studies are awaited.